# A Sustainable Methodology Using Lean and Smart Manufacturing for the Cleaner Production of Shop Floor Management in Industry 4.0

Varun Tripathi [1], Somnath Chattopadhyaya [2], Alok Kumar Mukhopadhyay [3], Shubham Sharma [4,5,*], Changhe Li [6] and Gianpaolo Di Bona [7,*]

1   Department of Mechanical Engineering, Accurate Institute of Management & Technology, Greater Noida 201306, India; varuncim@gmail.com
2   Department of Mechanical Engineering, Indian Institute of Technology (ISM), Dhanbad 826004, India; somnathchattopadhyaya@iitism.ac.in
3   Department of Mining Machinery Engineering, Indian Institute of Technology (ISM), Dhanbad 826004, India; akm_emm@yahoo.co.in
4   Department of Mechanical Engineering, IK Gujral Punjab Technical University, Main Campus-Kapurthala, Kapurthala 144603, India
5   Department of Mechanical Engineering, University Centre for Research and Development, Chandigarh University, Mohali 140413, India
6   School of Mechanical and Automotive Engineering, Qingdao University of Technology, Qingdao 266520, China; sy_lichanghe@163.com
7   Department of Civil and Industrial Engineering, University of Cassino and Southern Lazio, 03043 Cassino, Italy
*   Correspondence: shubham543sharma@gmail.com or shubhamsharmacsirclri@gmail.com (S.S.); dibona@unicas.it (G.D.B.)

**Abstract:** The production management system in Industry 4.0 is emphasizes the improvement of productivity within limited constraints by sustainable production planning models. To accomplish this, several approaches are used which include lean manufacturing, kaizen, smart manufacturing, flexible manufacturing systems, cyber–physical systems, artificial intelligence, and the industrial Internet of Things in the present scenario. These approaches are used for operations management in industries, and specifically productivity maximization with cleaner shop floor environmental management, and issues such as worker safety and product quality. The present research aimed to develop a methodology for cleaner production management using lean and smart manufacturing in industry 4.0. The developed methodology would able to enhance productivity within restricted resources in the production system. The developed methodology was validated by production enhancement achieved in two case study investigations within the automobile manufacturing industry and a mining machinery assembly unit. The results reveal that the developed methodology could provide a sustainable production system and problem-solving that are key to controlling production shop floor management in the context of industry 4.0. It is also capable of enhancing the productivity level within limited constraints. The novelty of the present research lies in the fact that this type of methodology, which has been developed for the first time, helps the industry individual to enhance production in Industry 4.0 within confined assets by the elimination of several problems encountered in shop floor management. Therefore, the authors of the present study strongly believe that the developed methodology would be beneficial for industry individuals to enhance shop floor management within constraints in industry 4.0.

**Keywords:** lean manufacturing; smart manufacturing; cyber–physical system; flexible manufacturing system; artificial intelligence; industry 4.0 technologies

## 1. Introduction

The revenue of Industry 4.0 is highly influenced by the problems that occur in the production system. These problems make it very difficult to manage the operations according to the economic conditions of the industries. That is why in recent years, production management systems have used some methods which include lean manufacturing, kaizen, smart manufacturing, a flexible manufacturing system, a cyber–physical system, artificial intelligence, and the industrial Internet of Things in the present scenario [1,2]. However, all of these techniques have shown their usefulness by improving productivity. However, lean manufacturing and smart manufacturing have proven their suitability for sustainable production systems in the context of Industry 4.0 [3,4]. Lean is a highly preferred approach in industries because it can be applied to improve production conditions within restricted resources [5–7]. The production conditions illustrate which type of problem is responsible for the present condition of the production which includes a higher production time, a higher inventory level, poor quality, an excess of manufacturing defects, mismanagement of types of equipment, and a lack of work experience [8–10]. All these problems can be eliminated simultaneously by using lean with smart manufacturing within the financial conditions, which cannot happen simultaneously by the implementation of other techniques [11]. Smart manufacturing works as a booster for shop floor management and improves the effectiveness of the overall production management system [12,13]. The main objective of lean and smart manufacturing in the context of Industry 4.0 is illustrated in Figure 1.

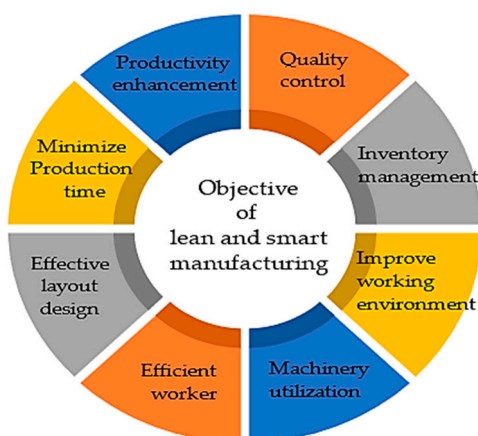

**Figure 1.** Objective of lean and smart manufacturing.

The implementation of hybrid lean and smart manufacturing has been documented in many previous studies, as discussed in the next paragraph. Shahin et al. [14] investigated how the operational performance of industries was affected by the implementation of lean and Industry 4.0 technologies. The study also discussed potential and existing lean technologies enabled by Industry 4.0 technologies, big data, virtual reality, wireless networks, and cloud computing. Finally, the study presented a decision support system for integrating kanban and cloud computing.

Bauer et al. [15] proposed two integrated teaching methods through the introduction of Industry 4.0 into the program of the learning factory. The concept was implemented in iwb's learning factory. The iwb's Learning Factory taught the methods and principles of sustainable lean production and hence provided a reality–conform smart manufacturing environment for the assembly of planetary gearboxes. The results showed the effectiveness of the above functional approach for the further development of smart teaching concepts in the context of Industry 4.0. Mora et al. [16] developed a model that connects lean and smart manufacturing to obtain goals in smart factories. The study was conducted in a medium-size cooling unit in relation to dispensing systems for cold drinks. The study showed that the developed model was able to achieve improvement by using a system that monitored the performance of workers and updated competencies in skills.

The lean manufacturing concept has been implemented in the last decades. The trend of smart manufacturing has increased in recent research because it can provide a sustainable production system [17]. It has been observed that the effectiveness of lean can be enhanced by an add on to the smart manufacturing concept. Lean with smart manufacturing improves production planning and focuses on sustaining product quality and diversity at a competitive cost [18,19].

In a vast literature review, it was observed that many researchers have proposed a number of approaches for production enhancement. They considered the complexities of the working environment including non-standardization working, product variety, worker skills, and work in process [20]. Singh and Singh [21] improved production by identifying the non-value-added activities by analyzing production conditions on the shop floor. It was found that the identified non-value-added activities were found among the different manufacturing processes, and these sluggish activities resulted in an increment in LT, CT, waiting time, and poor quality. The results revealed that the lean principle could significantly reduce the shop floor's WIP CT, and LT inventory.

Seth et al. [22] investigated the implementation of the lean concept in a complex environment by modifying value stream mapping. In the study, a case study was carried out in a power transformer manufacturing industry, and the work plan was drawn using a systematic questioning technique and Gemba walks. The results found that a VSM application-based lean message was similar for both a simple and complex environment. The results also showed that the root cause of application complexities includes non-compliance in relation to VSM assumptions and micro-concepts. Vinodh et al. [23] implemented VSM in a camshaft manufacturing organization. First, they developed the present state map after the necessary visualizations and calculations. After that, they identified various non-value-added activities and proposed the future state map. The results showed that VSM could be practically implemented in manufacturing industrial scenarios to improve leanness.

Andrade et al. [24] implemented lean in the auto parts industry. The study showed that VSM with a simulation is an efficient decision-making approach to obtain improvements in the production processes. Cheng et al. [25] integrated lean production and radio frequency identification technology to improve the effectiveness of warehouse management. In the study, VSM was designed to draw present and future state maps containing material flow, information, and time. The results showed that the total operation time from the start stage to the modified stage with only lean saved up to 79%. With the further integration of radio frequency identification to lean, they reduced the total operation time by 87%.

Sahoo et al. [26] implemented the lean principle in the forging industry. The study provided a strategy to implement the lean principle and Taguchi's method. The results showed a reduction in set-up time and inventory. Das et al. [27] implemented lean manufacturing to improve the productivity of air conditioning coil manufacturing. In the study, lean tools including kaizen, single-minute exchange of die, and VSM were implemented. The results showed a 67% improvement in the setup time and a 77% improvement in productivity in coil manufacturing and reduced the WIP inventory. Li et al. [28] presented a conceptual framework for the implementation of lean smart manufacturing. The authors observed how the bicycle industry located in Taiwan implemented lean smart manufacturing. The several methods used included visits, annual reports, and interviews that were performed to analyse information about the management system. The results showed the importance of combining the lean concept with Industry 4.0 and the set up of a smart manufacturing system. Abubakr et al. [29] discussed the integration of sustainable smart manufacturing performance and the challenges faced by the management system. The authors presented a comprehensive study and addressed two aspects including benefits for the implementation of sustainable manufacturing and the challenges faced by the management system to establish sustainable smart manufacturing. The results revealed that sustainable smart manufacturing would improve the environment quality in the management system in the future.

In the detailed literature review, it was observed that few notable works reported on the implementation of the lean concept with smart manufacturing. It was observed that the production management system is facing uncertain conditions in operational management including various conditions such as continuous changes in customer demand, longer downtime, absentees, lack of resources, congestion on the shop floor due to unavoidable reasons, etc. [30,31]. It was also observed that a higher level of improvement cannot be achieved by implementing lean manufacturing alone without a strategy. Therefore, the objective of the present study is to develop a sustainable methodology to control uncertain conditions through lean and smart manufacturing in industry 4.0.

Keeping all these facts in mind, in the present study the authors validated the developed methodology by productivity enhancement that was achieved in two case studies performed at an automobile manufacturer and an earthmoving machinery manufacturer. The developed methodology would help industry individuals to make a robust and cleaner working environment. The novelty of this work lies in the fact that the proposed methodology has been developed for the first time and this methodology helps the industry individual to enhance production in Industry 4.0 within confined assets by the elimination of several problems encountered in the shop floor management. Therefore, the authors of the present research believe that the developed methodology would be beneficial for production managers to enhance shop floor management within the available resources in industry 4.0.

*Lean and Smart Manufacturing: Conceptual Framework, Main Components, Their Objectives, and Their Respective Implementation in Industry 4.0*

In the present competitive industrial scenario, lean and smart manufacturing is an essential need of production management teams. The lean principle is used to maximize production by the elimination of waste and the smart concept helps to control operations management on the shop floor using modern technologies. Smart manufacturing is used to control operations management by continuously analysing operational performance [32]. The production management teams follow a conceptual framework to efficiently implement lean and smart concepts on the shop floor. The framework involves four steps: observation, analysis, improvement, and verification. The framework helps production management teams in the decision-making phase for operation management on the shop floor. In the present scenario, the lean and smart concept was proven to be a booster for production management teams to enhance operational excellence in all types of shop floor management. The main objectives of lean and smart manufacturing components are to improve workflow on the shop floor by identifying waste. The main components include total productive maintenance, value stream mapping, setup reduction, continuous improvement, internet of things, an artificial neural network, and an asset tracking system. It has been found that production management teams succeed in establishing a positive working environment on the shop floor using smart and lean manufacturing in industry 4.0. The same study was performed by Lee et al. [33] and they discussed the readiness of smart predictive informatics tools to manage big data and trends of manufacturing service transformation in the big data environment. The present research has been carried out in a smart remote machinery maintenance system. The study revealed that the prediction of machine health could reduce machine downtime and the prognostics information could support the ERP system to optimize maintenance scheduling, manufacturing management, and machine safety. In addition, the industry's new trend could provide a better working environment and reduce costs.

## 2. Sustainable Operational Excellence on the Shop Floor and Industry 4.0

The selection of an appropriate approach for production planning plays a vital role in improving operational excellence on the shop floor. The production planning approach allows productivity enhancements to be obtained within available resources. In the Industry 4.0 era, advanced techniques are used to enhance the efficiency of the production planning

approach on the shop floor. The advanced techniques mainly include the internet of things, the cyber-physical system, artificial intelligence, machine learning, and digitalization. The advanced techniques help industry individuals understand the production processes on the shop floor and suggest precisely how the operational performance can improve by using an efficient action. Advance techniques work more efficiently and enhance production by eliminating waste, establishing an aesthetic working environment, and providing higher profitability. Similar work has been reported by Tao et al. [34] who developed a conceptual framework and discussed the role of big data in supporting smart manufacturing. The study was carried out in a silicon wafer production line. The study revealed that the research provided three perspectives on the contributions of smart manufacturing, including historical perspectives, development perspectives, and envisioning the future of data from a manufacturing perspective.

### 2.1. Implementing Lean Manufacturing and Industry 4.0 Techniques

The present work aims to develop a sustainable methodology that uses lean and smart manufacturing to enhance production by eliminating waste in industry 4.0. Lean manufacturing improves operational performance by enhancing the production plan and reducing waste by eliminating idle activities [35]. Waste is produced by unnecessary activities performed on the shop floor, and it can never add any positivity to the product [36]. So, management teams focus on developing a sustainable model for controlling operation management on the shop floor using a suitable approach with advanced techniques to eliminate waste in industry 4.0. It can be obtained by implementing lean and smart concepts with advanced techniques. The advanced techniques help collect production shop floor information more effectively. The advanced techniques help industry individuals analyze production planning and suggest an exact action plan to enhance operational excellence on the shop floor, including the Industry 4.0 environment. Similar work has been reported by Lee et al. [37] who developed a systematic framework using cyber–physical integration, digital twins of different perspectives of a shop floor design from unit level, and business applications all together into an end-to-end design solution. In addition, a time machine was proposed to virtually evaluate different designs based on their performance over time in the research work. The results of the study revealed that the developed framework could satisfy short and long-term business requirements and provide predictable and expected outcomes.

### 2.2. The Link between Lean Manufacturing and Industry 4.0 Techniques

The lean principle can positively enhance operational excellence, and Industry 4.0 techniques give an action plan for improvement in operational excellence within limited constraints. The developed methodology can improve productivity by eliminating waste and maximizing resource utilization within available resources. The present study proved that the Industry 4.0 concept supports the lean idea by overcoming the limitations of the shop floor planning approaches. The lean manufacturing principle focused on improving operation management by eliminating idle activities. At the same time, the Industry 4.0 concept uses advanced techniques to improve operational performance by implementing an efficient action plan. In industry 4.0, industry individuals focus on developing a sustainable monitoring system to control production activities on the shop floor within available resources by implementing a suitable production planning approach. The methodology developed in the present research work can fulfill the needs of industry individuals and enhance financial profitability within confined assets.

### 2.3. Recent Development of the Production Management System of Industry 4.0

Industry 4.0 is undergoing rapid development in the present industrial environment. Several advanced techniques, such as smart manufacturing, artificial intelligence, internet of things, and the cyber–physical system have been used in industries. These techniques are preferred in the present scenario because they help industry individuals to make the

management system smart and capable of addressing problems and challenges [38,39]. Through these techniques, real-time production processes data can be obtained and rapid and precise decision-making can be facilitated. Lean has been revolutionized to address the production problems encountered in operations management and the integration of lean with smart manufacturing has emerged [40,41]. Smart manufacturing implementation with lean concept helps the management system to provide an intelligent decision-making system in industry 4.0.

Previous research has identified the perspectives, problem-solving keys, and applications for Industry 4.0 and has examined the challenges and problems faced in operation management. To alleviate these challenges, several techniques have been used that include smart manufacturing, the internet of things, and artificial intelligence. These techniques have been used to enhance the effectiveness of traditional approaches. However, no research has established a sustainable methodology of the lean concept with smart manufacturing in Industry 4.0. This is the motivation for the present research; this study develops a sustainable methodology for Industry 4.0 with lean and smart manufacturing. Industry 4.0 emphasizes the creation of smart manufacturing systems for the effectiveness of the lean concept within restricted resources. Smart manufacturing with lean can be defined as an automated system with the physical appearance of the management system. It responds in real-time to meet customer demands and conditions within financial conditions. Figure 2 presents the advancements in production management in industry 4.0.

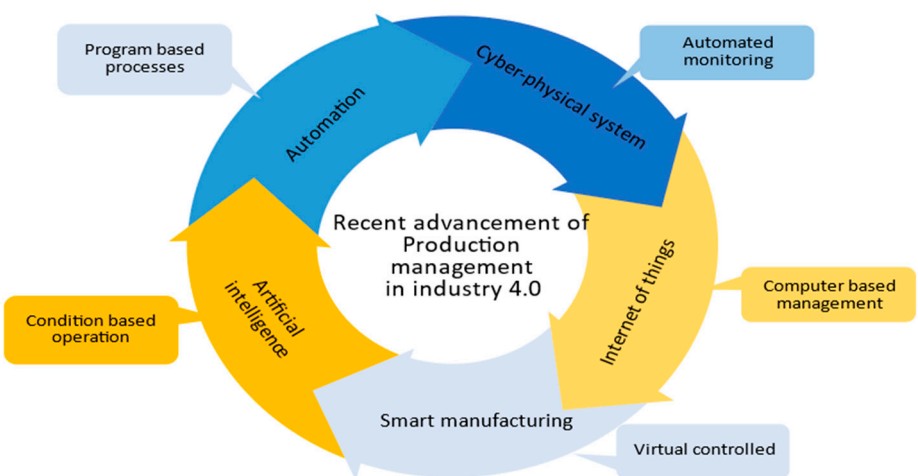

**Figure 2.** Recent advancements in industry 4.0.

Similar work has been reported by Kusiak et al. [42] who developed a data-driven approach to innovation and evaluated the innovativeness of the design of new products and services and planned introductions of design changes. The collected data and requirements were analyzed and refined by tools and human resources in the study. The study showed that the developed approach could set a new paradigm in innovation.

## 3. Research Methodology

In the present study, a methodology was prepared by thoroughly analysing the problems encountered in previous research articles. The developed methodology helps production management teams to establish sustainable production planning on the shop floor and also provides production enhancement within available resources. This statement for the developed methodology has been verified by achieving production enhancement in two different shop-floor conditions.

Industry 4.0 faces a wide range of challenges and problems from production planning design to logistics. Lean is a basic concept in present manufacturing systems and has been modified to suit working conditions [43,44]. Smart manufacturing is based on industrial needs that target individualized customer satisfaction in terms of product quality and de-

livery time [45,46]. The present research objective is to develop a sustainable methodology to enhance productivity through lean and smart manufacturing. The research provides a problem-solving key to industry persons to design production planning, logistics, condition monitoring, and scheduling. The developed methodology has been implemented in two case studies in industry A and industry B. The result of both studies validated the effectiveness of the developed methodology by obtaining productivity enhancement.

The proposed methodology consists of five phases. In the first phase, the performance of all the activities involved in production on the shop floor is observed. All performances are collected from one or more sources which involves consultation with workers, and management, study, industry records, questionnaire, interviews, etc. The second phase involves the categorization of all the collected documents according to the requirement of the production system. This categorization is performed on the basis of the type of production, type of layout, skill requirement, machinery requirement, and the number of operators. In the third phase, the present workflow is prepared from the acquired data from the sources as discussed in the first phase. These parameters include the sequence of operations, type of layout, number of the operator on each process, number of machineries per activity, time of operations. In the fourth phase, production processes are improved by the elimination of non-productive activities. This improvement is obtained by the calculation and analysis of the shop floor parameters in the previous phase. Finally, all the processes are evaluated in the fifth phase, and the results are compared with the predictable specification of a product provided by the industry's management system. It was also decided that if the product passes all standard specifications, the proposed map is implemented, otherwise the fourth phase is returned to.

In the proposed methodology, the complete workflow is shown by the flowchart, and the stations or activities of production going on the shop floor that are facing problems are investigated. Thereafter, appropriate techniques are implemented to eliminate them. Improper workflow arrangement, lack of resources, unskilled workers, incorrect work setup, an excess number of stations, and non-essential product handling are some crucial issues. To overcome these problems, a methodology is proposed in the present research work. The flow chart of the proposed methodology is shown in Figure 3.

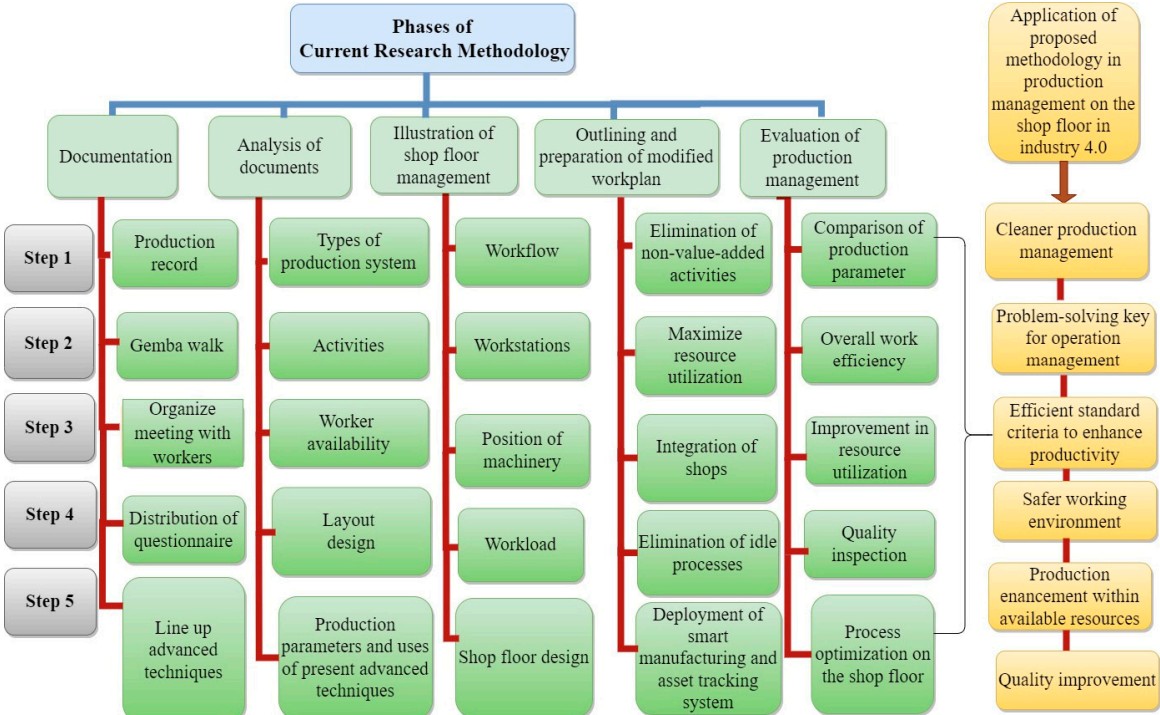

**Figure 3.** Phases of the proposed methodology.

The proposed methodology, i.e., the integration of hybrid lean and smart manufacturing for Industry 4.0 work was carried out in two different industries, i.e., industry A which signifies the automobile industry, and industry B which relates to the earthmoving machinery industry. These are thoroughly explained in a detailed discussion in the following paragraphs.

## 4. Application of Lean and Smart Manufacturing in Industry A and Industry B: A Case Study to Enhance Productivity and Operational Performance

In this section, the proposed methodology is implemented with lean and smart manufacturing in industry A and industry B. Industry A is the automobile industry and Industry B is an earthmoving machinery assembly unit. All the non-value-added activities were identified in the previous section; that is, the integration of hybrid lean and smart manufacturing and to further cognizing major issues of the problem in production. The common characteristics and differences between Industry A and Industry B are described in Figures 4 and 5.

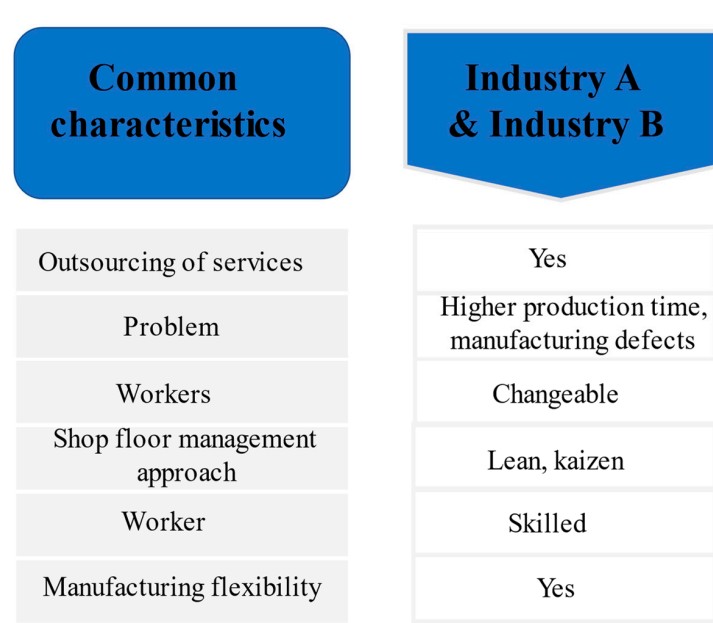

**Figure 4.** Description of common characteristics in both industries.

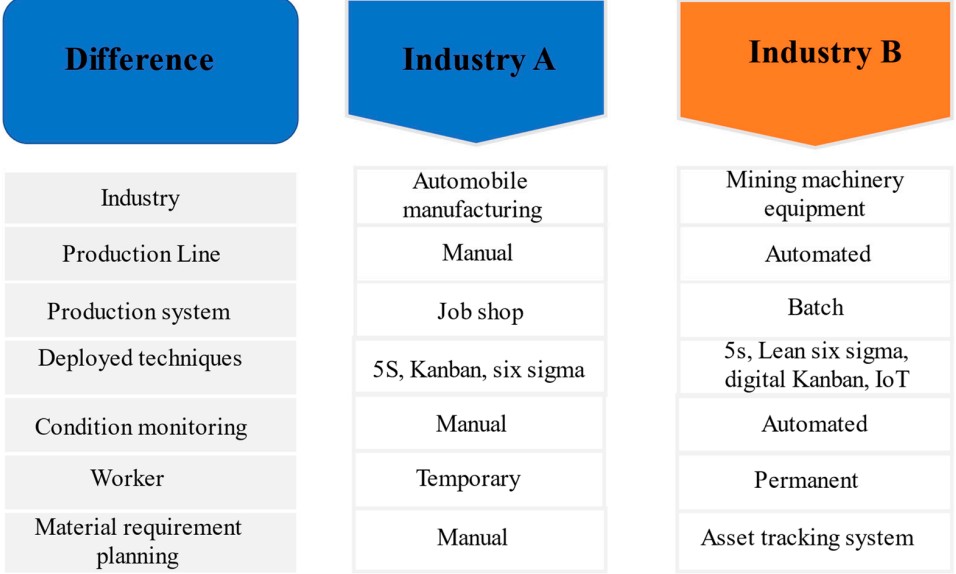

**Figure 5.** Differences between industry A and industry B.

### 4.1. Industry A: Automobile Industry

The main issue was poor productivity in the present industry which was found in relation to the fabrication station, where the setup of chassis manufacturing was found to be more complicated. This affected the overall performance of the production system significantly. The cylindrical rod pipe with a jig and fixture setup is used in the manufacturing of chassis whereas and metallic inert gas welding is used for fabrication. In the present case study, the defective production of chassis was found to be the main problem caused by placing the pipes in an inappropriate position prior to the welding operation. This improper arrangement arose due to a bumpy base. As a result, a sinuous shape was obtained in the chassis with weaker joints. To repair this, the joined pipes were straightened using heating and forming processes according to the needs of the production and then welding was performed. These types of incidents affect production and increase the chances of discontinuation in the workflow on the shop floor and result in poor quality with a relatively higher production time. In the present industry, the production is job-shop type and it depends on the customer demands. Thereby various processes were identified as bottleneck processes. This results in a lower quality, delayed orders, harassment of workers, a higher inventory time, a higher cost, a greater chance of machinery failure, loss in the image of industry, etc. To alleviate these problems, there is a need to identify all non-productive activities and bottleneck processes involved on the shop floor, for which the proposed methodology was implemented.

#### 4.1.1. Documentation

This case study was carried out at an automobile manufacturing industry located in India. All the data which were required for the case study were collected from the discussion with employees from observation, from the questionnaire, and the available company records. The following major data shown in Table 1 were collected from the shop floor.

**Table 1.** Observed data from the shop floor.

| S.N. | Data | Quantity/Amount |
|------|------|-----------------|
| 1. | Number of shifts | 2 |
| 2. | Working time | 480 min |
| 3. | Number of processes | 18 |
| 4. | Operating system | 2 |
| 5. | Number of workers | 8 |
| 6. | Planned downtime | 60 min |
| 8. | Automated machinery | Tungsten inert gas welding |
| 7. | Total working time | 960 min |

#### 4.1.2. Analysis of Documents

Analysis of documents means the categorization of all collected data according to the specification required for production condition in which production mode, types of machinery requirement, layout, number of operators, workplace requirement are the main specifications. The selected automobile industry is based on a process layout and based on a job-shop production mode system. The manufacturing system in the present industry follows a push system. In this case study, most of the operations were performed by operators one by one due to the insufficient number of operators. Various types of machinery are required for operations on the shop floor and this number may be varied according to production demand. Cutting, bending, pin marking, grinding, buffing, welding, and painting are some major machines which are used on the shop floor for the production of the vehicle.

### 4.1.3. Demonstration of Production

This industry receives orders on a monthly basis and releases a routine basis order to the management. Various operations are required in the manufacturing of vehicles including cutting, bending, pin marking, grinding, buffing, welding, and painting. The number of operators deployed to perform the operations on the shop floor depends on the requirements. The process flow observed in the present industry is shown in Figure 6.

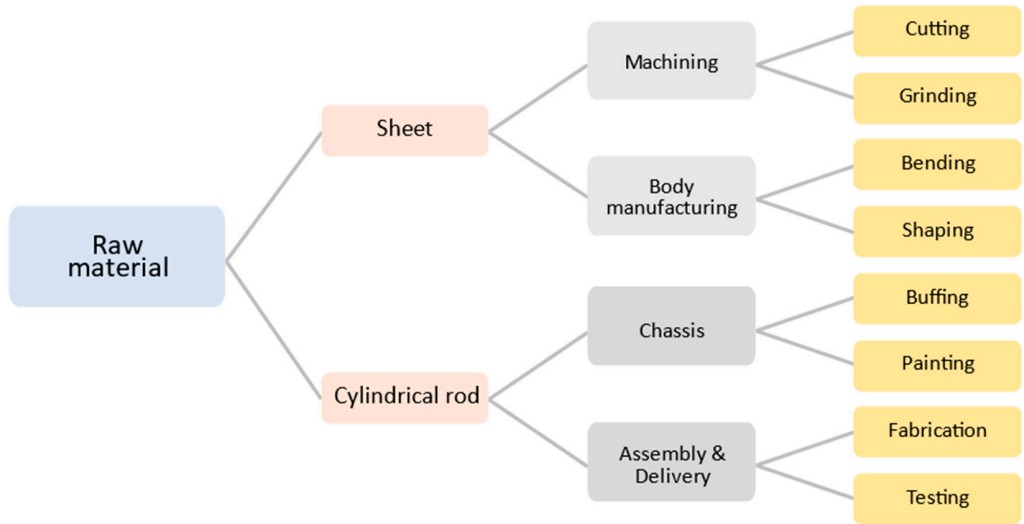

**Figure 6.** Process flow chart of the present shop floor.

### 4.1.4. Planning for a Modified Shop Floor

After an analysis of the performances of all the activities on the shop floor, the development of an efficient proposed state map with systematic planning was required. This was possible by selecting an optimized path for the processing of the product from the start point to the endpoint of the production system. Therefore, a new sequence of working with an upgraded position of welding on chassis is presented here to improve manufacturing. Figure 7 shows the proposed or improved sequence of operations involved on the shop floor.

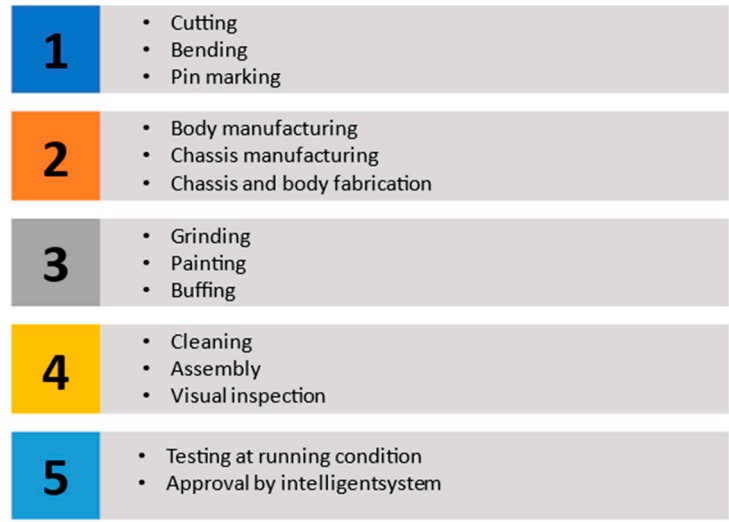

**Figure 7.** Proposed stepwise process flow chart.

### 4.1.5. Validation of Production Management

In the improved production management, it is suggested that the industry will operate in two shifts with 480 min/shift and 50 min/shift downtime with 15 operations. Table 2

shows the parameters and processes of the improved production management. Figure 8 illustrates the modified workflow for the proposed shop floor.

**Table 2.** Processes and calculated time for the modified shop floor.

| S.No. | Process | Available Time (min) | Uptime (%) | Number of Workers | Changeover Time (min) | Cycle Time (min) |
|---|---|---|---|---|---|---|
| 1. | Sheet and Pipe cutting | 860 | 99.77 | 2 | 2 | 20 |
| 2. | Sheet and Pipe bending | 860 | 99.77 | 2 | 2 | 20 |
| 3. | Pin marking | 860 | 99.42 | 2 | 5 | 10 |
| 4. | Chassis manufacturing | 860 | 98.25 | 3 | 15 | 35 |
| 5. | Body manufacturing | 860 | 98.25 | 2 | 15 | 25 |
| 6. | Grinding | 860 | 97.67 | 2 | 20 | 25 |
| 7. | Shaping | 860 | 98.84 | 2 | 10 | 20 |
| 8. | Chassis and Body fabrication | 860 | 98.25 | 3 | 15 | 25 |
| 9. | Buffing | 860 | 98.84 | 1 | 10 | 20 |
| 10. | Powder coating | 860 | 99.42 | 2 | 5 | 15 |
| 11. | Oven | 860 | 94.19 | 1 | 50 | 100 |
| 12. | Cleaning | 860 | 98.84 | 2 | 10 | 15 |
| 13. | Assembly | 860 | 98.84 | 3 | 10 | 20 |
| 14. | Visual inspection | 860 | 99.42 | 1 | 5 | 15 |
| 15. | Testing | 860 | 98.84 | 2 | 10 | 45 |

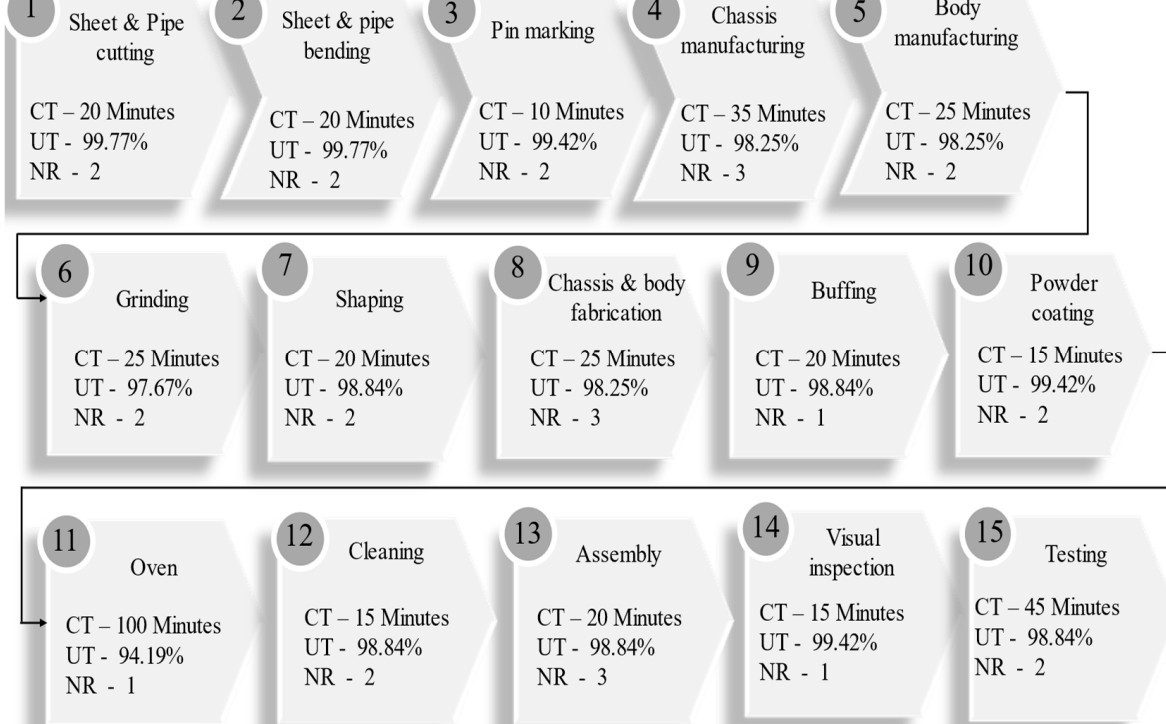

**Figure 8.** Modified workflow for the proposed shop floor.

In the improved map, all activities are in a systematic sequence with minimum utilization of available resources, minimum waiting time, minimum inventory, minimum non-productive activities, etc. It is also suggested that the proposed map may be implemented in the production system and may provide results in the form of a higher production rate with minimum cost. Table 3 shows an analysis of the parameters of the observed system and improved system on the shop floor in the industry.

**Table 3.** Analysis of the improvement achieved by the traditional and proposed methodology.

| S.No. | Parameters | Traditional Methodology | Proposed Methodology | Improvement |
|-------|-----------|------------------------|---------------------|-------------|
| 1. | Takt time | 280 min | 215 min | 65 min |
| 2. | Lead time | 840 min | 530 min | 310 min |
| 3. | Uptime | 77.44% | 80.51% | 3.07% |
| 4. | Numer of products/day | 3 | 4 | 1 |
| 5. | Workers skill level | Low-level skill | updated skill level and multi-tasking | 5 multitasking workers and up-gradation in 3 workers |

### 4.2. Industry B: Earthmoving Machinery

The study was carried out in a skid steer loader manufacturing in India. A skid steer loader is a leading earthmoving machine that uses cutting-edge technology. This machinery provides safety, easy maintenance, reliability, and low cost of ownership. It can survive the harsh conditions and has proven its suitability in any situation. During the observation, it was found that the present industry is facing challenges and problems regarding the higher cost due to higher production lead time. This is a serious issue affecting the ability to meet the customer's need in terms of delivery time. Some challenges and problems have been also found in the assembly, painting, and fabrication shops such as the unnecessary movement between workstations, uncertainty in equipment position, and inexperienced workers. These problems significantly affect the financial condition of the industry. Thereby in the present research, a sustainable methodology for production enhancement has been developed and it has been completed by the elimination of the idle activities.

#### 4.2.1. Documentation

The production data were collected by a Gemba walk and discussions with the supervisor. Table 4 shows the production information of the present production planning.

**Table 4.** Observed data of the shop floor.

| S.No. | Data | Quantity |
|-------|------|----------|
| 1. | Number of shifts | 1 |
| 2. | Working time | 580 min |
| 3. | Downtime | 40 min |
| 4. | Operating system | 3 |
| 5. | Number of workers | 46 |
| 6. | Available time | 540 min |
| 8. | Automated machinery | Profile cutting |

#### 4.2.2. Analysis of Documents

The observed production conditions were analyzed to gain an understanding of the real conditions of the management system. Some important production data taken into account for the analysis included working time/shift = 580 min, the number of shifts = 1, downtime = 60 min, and available time/shift = 540 min. It was observed in the analysis that the production system has six workstations that include assembly, painting, quality, fabrication, hot testing, and profile cutting.

#### 4.2.3. Demonstration of Production System

In the present industry, a job-shop production system is implemented and production is completed on regular basis. The industry used to produce the requirements of the customer. A Gemba walk and visual inspection were used to observe the real production conditions at the workstations. Figure 9 shows the present production planning found on the shop floor.

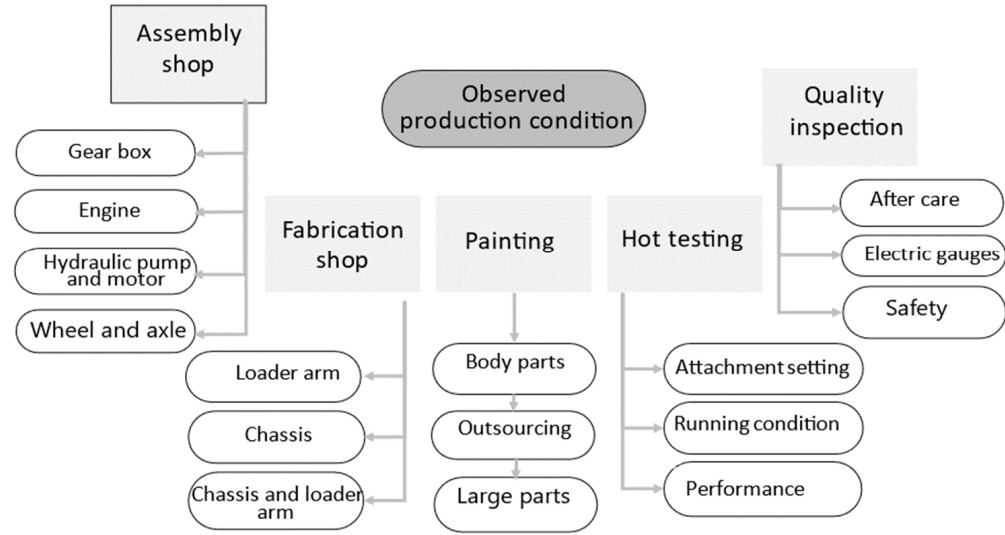

**Figure 9.** Present production planning found in industry A.

### 4.2.4. Planning for Modified Production Shop Floor Management

In the present industry, several non-value-added activities were identified by the implementation of the proposed methodology. After examining the production conditions, a revised production plan was developed by the authors. The modified production planning was validated by a discussion with the supervisor and industry persons. A new layout was developed to optimize the production processes and improve resource utilization. Modified production planning helps the management team to control uncertain conditions. Figure 10 shows the modified production planning.

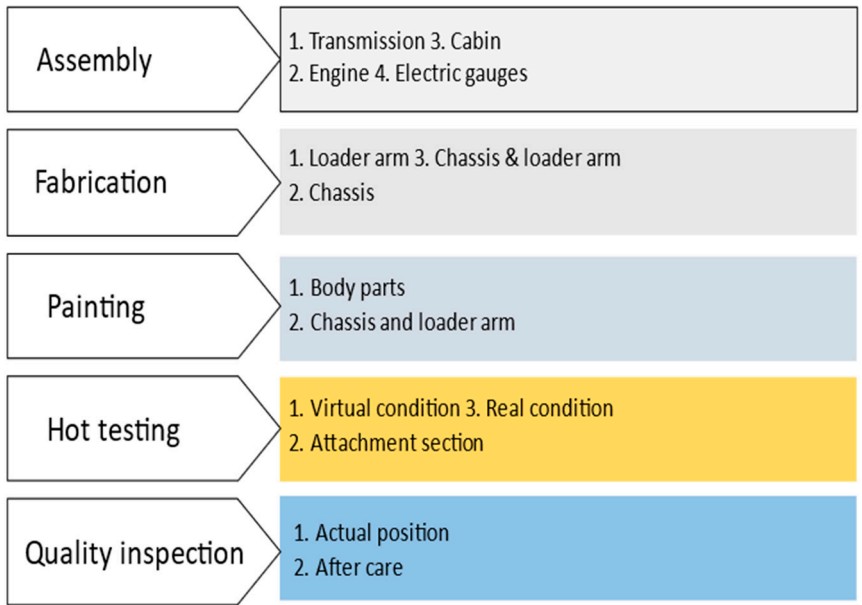

**Figure 10.** Modified production shop floor management.

### 4.2.5. Validation of Production Management

The validation of modified production planning was completed by comparing the values of the production parameters. In the modified production planning, it was suggested that the industry could operate in two shifts with 540 min of working time and 60 min of lunch break time. The production planning included 15 production processes and 42 workers with 4 reserved workers. Table 5 demonstrates the value of the parameters obtained by the modified production planning. Figure 11 shows the modified workflow for the proposed shop floor.

**Table 5.** Production parameters obtained in modified shop floor.

| S.N. | Process | Available Time (min) | Uptime (%) | Number of Workers | Changeover Time (min) | Cycle Time (min) |
|---|---|---|---|---|---|---|
| 1 | Gear box and Propeller shaft assembly | 520 | 97.11 | 4 | 15 | 120 |
| 2 | Axle and wheel assembly | 520 | 96.15 | 4 | 20 | 90 |
| 3 | Chassis manufacturing | 520 | 94.23 | 4 | 30 | 150 |
| 4 | Manufacturing of loader arm | 520 | 95.19 | 3 | 25 | 120 |
| 5 | Chassis and loader arm fabrication | 520 | 93.26 | 5 | 35 | 160 |
| 6 | Painting | 520 | 99.04 | 3 | 5 | 2150 |
| 7 | Engine assembly | 520 | 95.19 | 3 | 25 | 65 |
| 8 | Hydraulic pump and motor assembly | 520 | 98.07 | 2 | 10 | 60 |
| 9 | Roll off | 520 | 97.11 | 3 | 15 | 35 |
| 10 | Hot testing | 520 | 90.38 | 5 | 50 | 2940 |
| 11 | Cabin installment and Electric gauge assembly | 520 | 96.15 | 3 | 20 | 270 |
| 12 | Quality inspection | 520 | 99.04 | 3 | 5 | 105 |

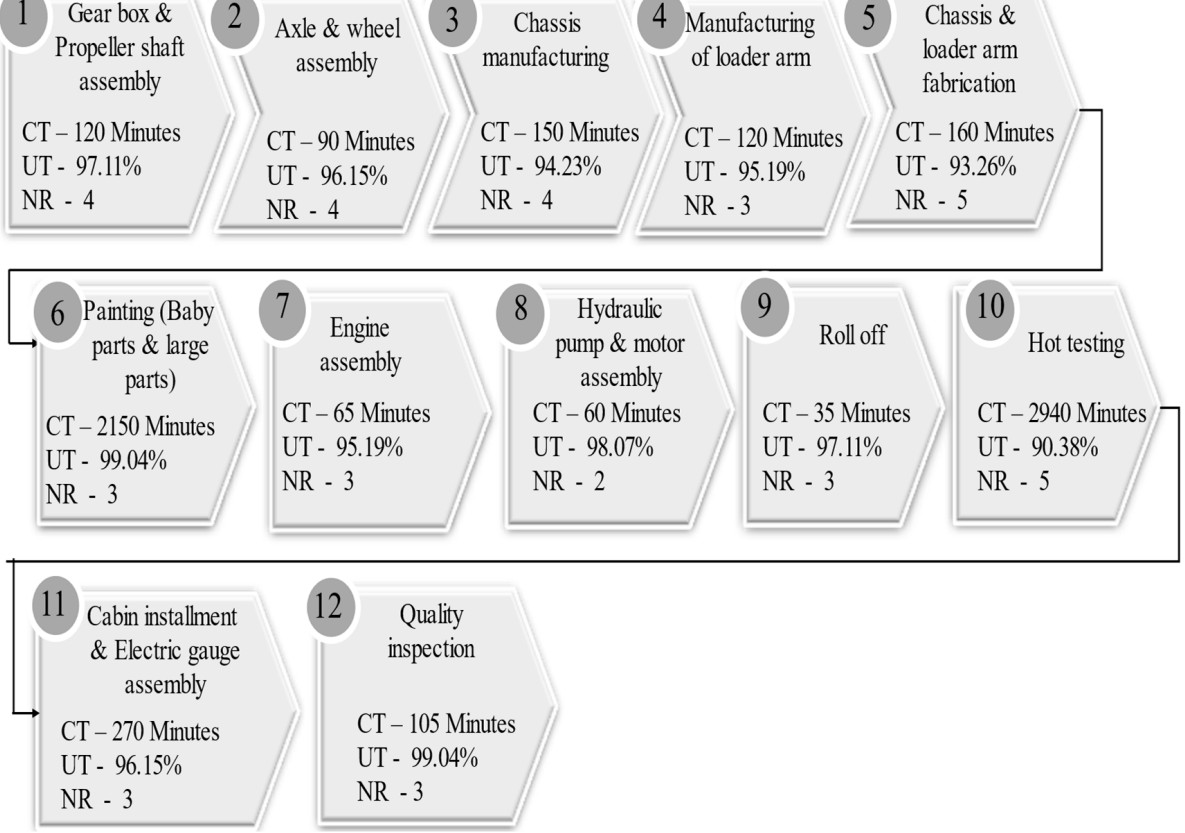

**Figure 11.** Modified workflow for the proposed shop floor.

For the new production parameter values in the modified production planning, 12 processes were taken into consideration. In the revised plan, 4 personnel were kept in backup which helps to reduce the work pressure on the people of the industry. It was proposed

that the reserve workers would be kept to cope with the uncertain conditions encountered in production. Table 6 shows an analysis of the parameters of the observed system and the improved system on the shop floor in the industry.

**Table 6.** Improvements in the various parameters of the shop floor.

| S.N. | Parameters | Traditional Methodology | Proposed Methodology | Improvement (min) |
|------|-----------|------------------------|---------------------|-------------------|
| 1. | Takt time | 125 min | 105 min | 20 min |
| 2. | Changeover time | 400 min | 275 min | 125 min |
| 3. | Lead time | 7270 min | 6895 min | 375 min |
| 4. | Idle time | 425 min | 250 min | 175 min |
| 5. | Uptime | 45.71% | 58.37% | 12.66% |

## 5. Results and Discussions

It was observed that the developed methodology proved to be efficient for obtaining sustainable and cleaner production shop floor management using lean and smart manufacturing. The developed methodology can eliminate non-productive activities efficiently in both industries, as well as improve the financial condition within restricted resources. Different modes have been used to collect data in industry A as well as industry B. The production data helps in understanding the accurate workflow condition. The workflow activities have been analyzed from the start production process to the end production process, and that has effectively improved productivity and overall performance. In the study, a visual inspection of the workstations identified key non-value-added activities, including separate sheet and pipe bending processes, separate sheets, pipe cutting processes, and improper welding steps which were found to be involved in Industry A, an, in Industry B, the key non-value-added activities included a lack of machinery and the additional number of inspection sections. Table 7 contains the problems of the shop floor and the actions taken to ensure improvement in production in industry A. Table 8 contains the problems of the shop floor and actions taken to ensure improvement in the production management system in industry B.

Through a Gemba's walk of the production shop floor and an evaluation of the production processes, it was observed that some activities were performed in an improper manner and were responsible for the poor product quality with a higher production lead time in both case studies. Tables 9 and 10 show the required implementation according to the production management system factors of both industry A and B. Y denotes the present relevant factor or the factor that may be applied, whereas N represents the absence of the factor or a lack of need to apply the factor. Related work has been reported by Reyes et al. [47] who developed a conceptual model that integrated lean manufacturing and Industry 4.0 technologies to reduce waste and cost in the context of a lean supply chain context. The presented conceptual model was validated with a case study on a footwear company. The presented model established structured relations among the agile, lean, resilient, sustainable, and flexible paradigm to the improved supply chain through Industry 4.0 enabling technologies. The results showed that the developed model can help decision-makers to improve the management and planning of digital supply chain production processes.

**Table 7.** Problem, Action, and Result / Results obtained by action taken against identified problems in industry A.

| S.No. | Present State Process | Problems | Actions |
|---|---|---|---|
| 1. | Sheet cutting | | |
| 2. | CR pipe cutting | Unnecessary movement between workstations. | Both cutting processes are performed on one workstation. |
| 3. | MS pipe cutting | | |
| 4. | Sheet bending | Unnecessary movement between workstations. | Both bending processes are performed at a single station. |
| 5. | Pipe bending | | |
| 6. | Pin marking machine | Manual operating system. | Provide computer-controlled machinery with a smart intelligence system. |
| 7. | Chassis manufacturing | Fabrication performed without support on ground resulting in various defects. | Using advanced welding processes with a permanent base. |
| 8. | Chassis grinding | Lack of equipment. | Provide a setup for the condition-based monitoring system. |
| 9. | Shaping | Lack of workers. | Improve workload plan. |
| 10. | Body manufacturing | Manual operation and equipment. | Use advanced machinery with the automation concept. |
| 11. | Body grinding | Lack of machinery. | Improve production planning. |
| 12. | Chassis and Body fabrication | Improper alignment due to lack of machinery. | Use computer-controlled equipment for alignment. |
| 13. | Buffing | No problem seen. | No action required. |
| 14. | Powder coating | Manual operation results in uneven coating layers. | Use automation concept with smart sensors. |
| 15. | Oven | Manual setting for temperature. | Use smart sensors for time and temperature settings. |
| 16. | Cleaning | No problem seen. | No action required. |
| 17. | Assembly | Defective output due to unskilled worker. | Organize training sessions. |
| 18. | Testing | Malfunction in the machinery due to faulty parts and errors in production planning. | Design production planning with optimum workflow. |
| 19. | Testing | Malfunction in the machinery due to faulty parts and errors in production planning. | Design production planning with optimum workflow. |

**Table 8.** Problem, Action, and Result/Results obtained by action taken against identified problem in industry B.

| S.N. | Production Process | Problem | Action |
|---|---|---|---|
| 1. | Gearbox and shaft assembly | Unnecessary movement and inspections. | Design new layout and production plan. |
| 2. | Manufacturing of loader arm | Longer setup time. | Increase the number of workers and improve the workplan. |
| 3. | Chassis and loader arm fabrication | Unnecessary movement between workstations. | Design a modified work plan. |
| 4. | Painting (Baby parts and large parts) | Outsourcing of services. | Provide advanced machinery for both parts within the plant. |
| 5. | Engine assembly | Cluttered equipment, and malfunctioning in the hoist system. | Use a condition-based monitoring system. |
| 6. | Hydraulic pump and motor assembly | Excess movement between workstations. | Both processes must be performed at a single station. |
| 7. | Roll-off | Unnecessary movement and document work. | Eliminate unnecessary activities. |
| 8. | Hot testing | More idle activities. | Modify production planning. |
| 9. | Cabin installment and Electric gauge assembly | Lack of communication gap between workers. | Organize meeting and training sessions. |
| 10. | Quality inspection | Lack of workload distribution. | Improvement in production planning. |

**Table 9.** Implemented factors according processes in industry A.

| S. No. | Factors | Sheet and Pipe Cutting | Sheet and Pipe Bending | Pin mark Machine | Chassis Manufacturing | Grinding | Shaping | Body Manufacturing | Chassis and Body Fabrication | Buffing | Powder Coating | Oven | Cleaning | Assembly | Testing |
|---|---|---|---|---|---|---|---|---|---|---|---|---|---|---|---|
| | | | | | | | | | | | | | | | |
| 1 | Bottleneck in operation | Y | Y | Y | Y | Y | N | Y | Y | N | N | N | N | Y | Y |
| 2 | External arrangement required | N | N | N | Y | N | N | N | Y | N | N | N | N | N | Y |
| 3 | Improvements required in machinery | N | N | N | Y | N | Y | N | Y | N | N | N | N | Y | Y |
| 4 | Improvements required in worker's skills | N | N | Y | Y | N | N | N | Y | N | N | N | N | N | Y |
| 5 | Automation required | N | N | Y | Y | N | N | Y | Y | N | Y | Y | N | Y | Y |

**Table 10.** Implemented factors according processes in industry B.

| S.N. | Requirement of Shop Floor | Gearbox and Propeller Shaft assembly | Manufacturing of Loader Arm | Axle and Wheel Assembly | Chassis Manufacturing | Chassis and Loader Arm Fabrication | Painting (Baby Parts and Large Parts) | Engine Assembly | Hydraulic Pump amd Motor Assembly | Roll-Off | Hot Testing | Cabin Instalment and Electric Gauge assembly | Quality Inspection |
|---|---|---|---|---|---|---|---|---|---|---|---|---|---|
| | | | | | | | | | | | | | |
| 1 | Bottleneck in operation | N | Y | Y | Y | Y | N | Y | Y | Y | Y | Y | Y |
| 2 | External arrangement required | N | Y | N | N | N | N | Y | N | Y | N | Y | N |
| 3 | Improvements required in machinery | Y | N | N | N | Y | Y | Y | N | N | N | N | N |
| 4 | Improvements required in worker's skills | N | N | N | N | N | N | N | N | N | N | Y | N |
| 5 | Automation required | Y | N | Y | Y | Y | N | Y | Y | N | N | Y | N |

The results of case study A reveal that takt time improved by 65 min, cycle time improved by 180 min, idle time improved by 60 min, changeover time improved by 27 min, and lead time improved by 310 min. The production system of industry B showed a reduction in cycle time by 655 min, takt time by 30 min, and non-value-added time by 520 min. Figure 12 shows a comparison between parameter improvement in industry A and industry B, respectively. A similar study has been reported by Aggarwal et al. [47] who investigated the relationship between smart and sustainable manufacturing practices. The study used the hypothesis modeling approach to link the top management commitments and manufacturing competitiveness with the smart and manufacturing practices. The data were collected by organizing a questionnaire survey at Indian manufacturing industries in the study. The study revealed how developing economies such as India could adopt sustainable and smart manufacturing practices to increase profit. Zheng et al. [48] developed a conceptual framework of a manufacturing system for industry 4.0. The present study reviewed vital technologies such as the internet of things, cyber–physical system, and big data analytics for Industry 4.0 smart manufacturing systems. The results of the study showed that the developed framework provided insights to industry individuals for implementing Industry 4.0. Similar work has been reported by Sony et al. [49] who developed an integrated lean manufacturing model and industry 4.0. The study integrated the vertical, horizontal, and end-to-end engineering integration models with lean manufacturing methodology. The study provides fifteen issues related to the integration of Industry 4.0 with lean manufacturing.

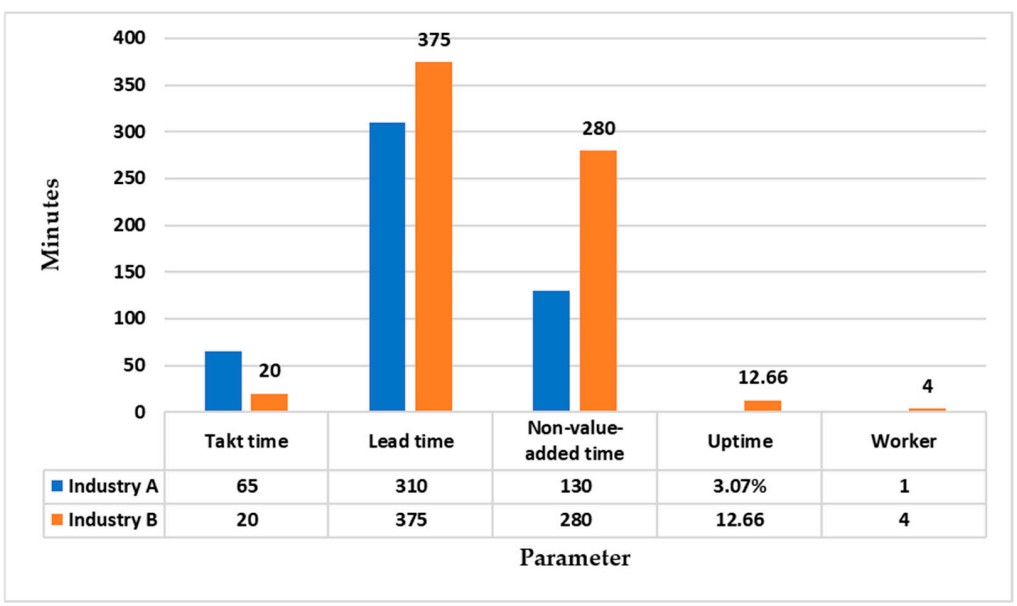

**Figure 12.** Improvement obtained by the developed methodology in parameters.

It was found in the results that the developed methodology is more efficient than the improvements obtained from the methodology developed in the previous research work by various researchers and from this the production management can be increased effectively. The associated results were proven by Amrani et al. [50] who implemented a lean manufacturing approach in the aerospace sector and found improvements in cycle time and defects by 43% and 66%. Caiado et al. [51] developed a model using a computational intelligence approach for supply chain management in industry 4.0. The presented model was evaluated by implementing it in a real case within the manufacturing industry. As a result, it was found that the developed model was able to provide a robust tool for digital readiness in manufacturing industries. Moreover, Thomas et al. [52] implemented lean six sigma to overcome the challenges faced in production management in an aerospace

company. The results showed that production time improved by 16.79% and financial profitability improved.

The proposed methodology has proven to be sustainable in achieving cleaner production management by the maximization of resources within confined assets. This statement has been proved by a comparison between the results found in the present research and previous research works. The proposed methodology has been developed for the first time and it helps industries to enhance production in Industry 4.0 within restricted resources by the elimination of several problems encountered in the shop floor management. Therefore, the authors of the present study believe that the current methodology would be beneficial for industry individuals to enhance shop floor management within limited constraints in industry 4.0.

## 6. Prospective Organizational Impact on Production Planning in Shop Floor

The developed methodology is based on a flexible manufacturing system concept. It is suitable for controlling variations in operating conditions on the shop floor by reducing resource consumption and enhancing financial profitability. Lean and smart manufacturing provide a strategic plan to management teams to identify the source of waste and to opt for an efficient way to achieve production enhancement within available resources. Furthermore, the lean concept is considered to be an efficient way of improving operational excellence by using advanced techniques in industry 4.0. Thus, it has been found that the lean approach and Industry 4.0 techniques with smart manufacturing are efficient and robust ways of controlling operational excellence on the shop floor.

## 7. Integration of a Sustainable Lean Methodology and a Digital Smart Manufacturing Approach for Enhancing Operational Excellence in Industry 4.0: A Comparative Analysis of the Current Research with the Previous Literature

From the previous studies in the literature, it is clear that management teams face problems in operation management on the shop floor in relation to the achievement of production enhancement within confined assets [3,7,11,20,31,41,47,53,54]. The problems are found in various forms and mainly include a higher downtime, ergonomics issues, a communication gap, a higher production lead time, and incompetent workers. The management team need to focus on the development of a methodology to remove these problems and to obtain the desired production enhancement. The results reported in the current research study were found to be an efficient way of obtaining an improved flexible production system within limited conditions. The production capacity was significantly improved by 85%, the manufacturing defects were drastically reduced to 95%, the production cost was dramatically reduced to 56%, and the machinery utilization was improved by 17%.

The current dynamic trend requires advanced techniques to facilitate effective production management on the shop floor. Therefore, production management teams emphasize the enhancement of operational excellence by developing a sustainable methodology using lean, smart, and digital approaches. A sustainable method helps manage the operational performance of production processes on the shop floor within available resources. The present study developed a sustainable methodology to use lean and smart manufacturing to enhance production within available resources. This study has shown how to control operations management on the shop floor effectively using lean and smart manufacturing in industry 4.0. The present research work supports the integration of a lean methodology and smart digital manufacturing to enhance operational excellence on the shop floor in industry 4.0. In the current scenario, the production management team wished to develop a sustainable methodology to overcome waste found on the shop floor using advanced techniques. The present work has helped to establish a positive work environment on the shop floor and enhanced production within available resources. Figure 13 describes the objectives and advantages of the current sustainable methodology in comparison with previous research studies.

| Objectives of current sustainable methodology for industry 4.0 | Performance assessment of technology developed in process parameter on production shop floor for industry 4.0 | Outputs obtained from previous research studies |
|---|---|---|
| • Develops a sustainable methodology for cleaner production system in industry 4.0. <br>• Establish an agile system to control working production condition on the shop floor. <br>• Provides a problem-solving key using industry 4.0 techniques to industry individuals for production management on the shop floor. <br>• Applicable in all types of working environment on different shop floors. <br>• | 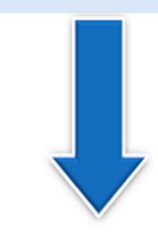 | • Enhancement in production rate on the shop floor within available resources. <br>• Reduced manufacturing defects by improvement in operation management on the shop floor. <br>• Provides approach for production management on the shop floor. <br>• Applicable in specific and limited condition on the shop floor. |

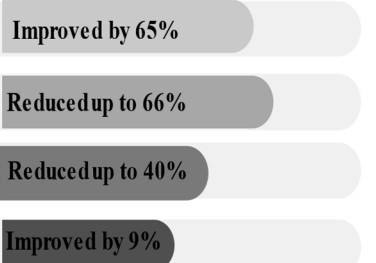

| | | |
|---|---|---|
| Improved by 85% | **Production capacity** | Improved by 65% |
| Reduced up to 95% | **Manufacturing defects** | Reduced up to 66% |
| Reduced up to 56% | **Production cost** | Reduced up to 40% |
| Improved by 17% | **Machinery utilization** | Improved by 9% |

**Figure 13.** Comparison of the outcomes between the current sustainable methodology and previous research studies.

## 8. Conclusions

In this article, the authors aimed to develop a model of lean and smart manufacturing to make a cleaner production system for Industry 4.0. The results of the case study support our premise that the lean and smart manufacturing concept could play an important role in Industry 4.0, and it also allows higher productivity enhancement to be obtained within restricted resources. The authors have summarized the findings obtained by the present research below:

i. The developed methodology was capable of improving both Industry A and B. After discussion and deliberation with industry individuals, it has been proven that implementing this methodology will effectively improve the production parameters, reducing the lead time in Industry A and B by 5.15% and 36.90%, reduce uptime in industry A and B by 3.07% and 12.66%, respectively, and improve production capacity in industry A and B by 33%.33 and 50% per day, respectively. The developed methodology can enhance operational excellence and financial profitability within restricted resources. The developed methodology would be beneficial to management teams by allowing them to control production processes on all types of shop floor including industry 4.0. The developed methodology was found to be sustainable in comparison to methodologies reported in previous research works.

ii. The present research aimed to develop a sustainable methodology using lean and smart manufacturing for cleaner shop floor management in industry 4.0. The developed methodology can enhance production on all production systems, including industry 4.0, within confined assets and available resources. The authors strongly believe that the developed methodology would help management teams in the decision-making phase in controlling production activities by implementing an exact production plan and action plan for production enhancement within restricted resources. Furthermore, the methodology helps improve the production processes' operational

performance by eliminating waste and the problems found on the shop floor, including industry 4.0.

iii. It was observed that the developed methodology can provide a sustainable and cleaner production system in Industry 4.0 and effectively control uncertain conditions on the production shop floor, including changes in customer demands, unavailability of resources, high downtime, and congestion on the shop floor.

iv. The authors highly recommend that industry individuals enhance the productivity and operational excellence of the respective shop floor in Industry 4.0 by using this novel hybrid framework of lean and smart manufacturing.

v. The authors of the present research work strongly believe that the developed methodology will help industry people to overcome the problems and challenges faced by the management systems in Industry 4.0 through the developed methodology.

## 9. Future Outlook

Based upon the current study, the authors have proposed a few recommendations as mentioned below:

i. The authors suggest that industry individuals could increase the effectiveness of the developed methodologies by using cyber–physical systems, artificial intelligence, and the industrial Internet of Things, and by integrating these with the lean concept, which together will provide a higher productivity.

ii. The research work highlighted the advancements obtained by smart manufacturing in Industry 4.0 and should inspire young researchers and industry individuals embarking on Industry 4.0 to extend this approach to improve productivity on the shop floors of various other industries.

**Author Contributions:** Data curation, C.L.; methodology, A.K.M. and S.S.; supervision, G.D.B.; validation, V.T.; writing—original draft, S.C. All authors have read and agreed to the published version of the manuscript.

**Funding:** This research received no specific grant from any funding agency in the public, commercial or not-for-profit sectors.

**Institutional Review Board Statement:** Not applicable.

**Informed Consent Statement:** Not applicable.

**Data Availability Statement:** The data presented in this study are available on request from the corresponding author.

**Conflicts of Interest:** The authors declare no conflict of interest.

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
