# Peer review of "A Sustainable Methodology Using Lean and Smart Manufacturing for the Cleaner Production of Shop Floor Management in Industry 4.0"

_mathematics, doi:10.3390/math10030347_

Round 1

Reviewer 1 Report

The topic of the paper is interesting and i enjoyed reading that. Here are my comments:

  • the authors combined the lean and smart manufacturing concept for sustainable production which is interesting. However, this combination is vaguely formulated. This is not clear if such approach is practiced by others. Figure 1 should be modified. Authors should specify the main elements of lean and smart manufacturing separately, explain the need to combine them and clearly formulate the combination.
  • The authors started the paper by explaining the Objective of lean and smart manufacturing and then focused on recent development in industry 4.4 management. Perhaps, they could focus on the lean+smart manufacturing after talking general about the industry 4.0 concept.
  • section 3 is not readily understandable. Figure 3 can be modified and authors need to clearly show how the methodology reflects the combination of lean and smart manufacturing. 
  • when talking about concepts such as industry 4.0, smart manufacturing, etc. referring to works of internationally recognized researchers in the fields are necessary. You may check the works of prof J. Lee, Prof A. Kusiak and many others... 
  • conclusion iii is not the direct results of this study but is more a general statement. if you plan to link your study to concepts such as AI, IoT, etc. please make a discussion on that and link it to literature. You can explain the concept of smart manufacturing vs. data driven smart manufacturing (refer to A review on smart data-driven manufacturing paper by A. Kusiak). Also, to really adopt such concepts delaing with challenges of data is necessary. In a recent "review on deep learning for machining and tool monitoring: methods, opportunities and challenges", both the opportunities and challenges about data-driven approach are explained. I recommend authors elaborate a discussion and then based on that, it is reasonable to reflect that in the conclusion.

  •    

Author Response

Dear Prof. (Dr.) Editor-in-chief,

Thank you for considering the manuscript entitled, “A sustainable methodology using lean and smart manufacturing for the cleaner production shop floor management in industry 4.0” (Mathematics - 1523846), for the publication in Mathematics journal. I am grateful to you and the reviewers for the valuable suggestions provided. I like to resubmit our revised version of the manuscript by adding response to all your comments. Below please find the answers and actions taken to address these comments. All the suggestions are incorporated and highlighted with the YELLOW COLOR in manuscript.

NOTE: All the necessary changes/added sentence has been shown by yellow colour.

The locations of these changes have been mentioned, where possible, in the action points that respond to each reviewers’ comments. Here are the responses to the reviewer comments:

AUTHOR RESPONSE TO REVIEWER AND EDITOR COMMENTS

Manuscript ID: Mathematics- 1523846

Paper title: A sustainable methodology using lean and smart manufacturing for the cleaner production shop floor management in industry 4.0

The manuscript has been thoroughly modified and improved the quality of the content to meet the standards of the Journal. All the suggestions made by the learned referees are included in the revised manuscript. We are extremely thankful to the referees & editor(s) for their constructive comments and appreciation.

Response to Reviewer’s Comments

The authors are grateful to the reviewers for their suggestions that have all contributed to improving the manuscript. Once again, the authors are extremely thankful for the observations and the comments of the reviewers. All the comments are appropriately addressed and now the quality of the article has been appreciably enhanced before the consideration for publications. The rebuttal file is enclosed indicating the revisions incorporated in the article as suggested. The revisions are carried out in yellow colour in the text of the manuscript for better visibility to the reviewers and as well as to the editor. We have made the modifications as per their suggestions in the revised manuscript and changes are also marked up using the “Yellow font colour” function.

All in all, the authors should thank the reviewers for their meticulous observations in reviewing the article. All the issues raised by the authors are appropriately addressed as stated in the following table.

Authors’ Reply to The Editor and Reviewer 1

Title: A sustainable methodology using lean and smart manufacturing for the cleaner production shop floor management in industry 4.0

Reply to Editor: Authors are thankful to the Editor for providing constructive comments on our manuscript. The authors’ have incorporated all the suggestions given by the editor and reviewer. The authors’ have formatted the manuscript along with references as per journal guidelines.

Manuscript ID: Mathematics - 1523846

Comment 1 - the authors combined the lean and smart manufacturing concept for sustainable production which is interesting. However, this combination is vaguely formulated. This is not clear if such approach is practiced by others. Figure 1 should be modified. Authors should specify the main elements of lean and smart manufacturing separately, explain the need to combine them and clearly formulate the combination.

Authors’ Reply – Authors are thankful to the reviewer for giving constructive comments. As per the valuable insights, the authors have thoroughly discussed the integration of the lean and smart manufacturing concept for sustainable production in industry 4.0 management with supportive recent scientific evident literature studies.

The correction has been shown on page 2, lines 66-83, page 4 (sub-section 1.1), lines 158-184, and page 5, lines 197-238.

Comment 2 The authors started the paper by explaining the Objective of lean and smart manufacturing and then focused on recent development in industry 4.4 management. Perhaps, they could focus on the lean+smart manufacturing after talking general about the industry 4.0 concept.

Authors’ Reply – The authors are thankful to the reviewer for giving constructive comments, as per the instructions, the authors have thoroughly discussed the combination of lean with smart manufacturing for the industry 4.0 concept, along with the recent developments on industry 4.0 management.

The correction has been shown on page 2, lines 66-83, page 3, lines 82-83, page 4 lines 158-184, page 5, lines 181-184, lines 204-221, page 6, lines 222-238.

Comment 3 section 3 is not readily understandable. Figure 3 can be modified and authors need to clearly show how the methodology reflects the combination of lean and smart manufacturing. 

Authors’ Reply – The authors are thankful to the reviewer for giving constructive comments, the section 3, i.e., Research Methodology, and figure 3 have been modified up to a fervent extent as per the standard Journal formatting.

The correction has been shown on page 7 lines 273-278, page 8, lines 314-317, and page 26, lines 576-609.

Comment 4 - when talking about concepts such as industry 4.0, smart manufacturing, etc. referring to works of internationally recognized researchers in the fields are necessary. You may check the works of prof J. Lee, Prof A. Kusiak and many others... 

Authors’ Reply – The authors are thankful to the reviewer for giving constructive comments, the reference has been included accordingly.

The suggested references have been cited and explained as shown on page 4 lines 177-180, page 5, lines 181-184, page 5, lines 198-202, lines 217-221, page 6, lines 222-223, page 7, lines 267-271.

Comment 5 - conclusion iii is not the direct results of this study but is more a general statement. if you plan to link your study to concepts such as AI, IoT, etc. please make a discussion on that and link it to literature. You can explain the concept of smart manufacturing vs. data driven smart manufacturing (refer to A review on smart data-driven manufacturing paper by A. Kusiak). Also, to really adopt such concepts delaing with challenges of data is necessary. In a recent "review on deep learning for machining and tool monitoring: methods, opportunities and challenges", both the opportunities and challenges about data-driven approach are explained. I recommend authors elaborate a discussion and then based on that, it is reasonable to reflect that in the conclusion.

Authors’ Reply – Authors are thankful to the reviewer for giving constructive comments. Based upon the aforementioned fruitful suggestions, the conclusions section has been ameliorated to a fierce extent.

Additionally, we have eradicated point number iii of the Conclusions section as by mistakenly, we have mentioned the information which undoubtedly is completely vague and we have framed that matter for the Future outlook section. Now, we have mentioned the same in the Future Scope.

As the authors have broadly used the Smart manufacturing concept without employing AI, IoT, etc., after emphasizing in-depth recent and more relevant literature studies. Additionally, the authors have extensively enumerated the concept with appropriately supportive prior literary findings as per the referee’s suggestions in the results & discussions and conclusions sections respectively.

The correction has been shown on page 25 lines 545-556, page 25-27, lines 566-609, and page 27-28, lines 617-637.

A Scientific Explanation of the obtained results has been refined and ameliorated up to a fervent extent. Results are enumerated, a methodology is utterly described, interpretation has been correlated with results and previous literature findings. The overall summary should indicate the progress of the research and the limitations. 

Note: All the necessary changes/added sentence has been shown in yellow colour.

Thank you very much in advance for taking your time in reviewing this manuscript.

Sincerely, we hope you will find our revision satisfactory.

Thanks, in anticipation.

Regards,

Shubham Sharma

Gianpaolo Di Bona

(Corresponding author)   

Reviewer 2 Report

First of all, I would like to congratulate the authors for their interesting research. This research can be helpful instrument for companies that have ambition to adopt parts of Industry 4.0 in their activities.

Comments:

  • Part 3 – It would be better, if you could give common features and differences of Industry a and Industry B into the table and you have to justification. Why did you choose this industries?
  • Part 4 and 4.1 and 4.2 – Is it the Industry A (B) or the company A (B)?
  • Table 1 and Table 4 – Do you think, that the sample is representative?
  • Part 4.1.2. – As you write, the system uses push system. How do you want to implement your proposed methodology with Industry 4.0 usage, while most of the parts that uses Industry 4.0 are based on the pull principle?
  • Table 2 and Table 5 – It would be better to replace these two tables with pictures – something like process maps.
  • Lines 420 – 433 – I think that is not right to give something like literature review into the section “Results and Discussions”. Please, find better place for this paragraph.

Author Response

Dear Prof. (Dr.) Editor-in-chief,

Thank you for considering the manuscript entitled, “A sustainable methodology using lean and smart manufacturing for the cleaner production shop floor management in industry 4.0” (Mathematics - 1523846), for the publication in Mathematics journal. I am grateful to you and the reviewers for the valuable suggestions provided. I like to resubmit our revised version of the manuscript by adding the response to all your comments. Below please find the answers and actions taken to address these comments. All the suggestions are incorporated and highlighted with the YELLOW COLOR in the manuscript.

NOTE: All the necessary changes/added sentence has been shown in yellow colour.

The locations of these changes have been mentioned, where possible, in the action points that respond to each reviewers’ comments. Here are the responses to the reviewer comments:

AUTHOR RESPONSE TO REVIEWER AND EDITOR COMMENTS

Manuscript ID: Mathematics 1523846

Paper title: A sustainable methodology using lean and smart manufacturing for the cleaner production shop floor management in industry 4.0

The manuscript has been thoroughly modified and improved the quality of the content to meet the standards of the Journal. All the suggestions made by the learned referees are included in the revised manuscript. We are extremely thankful to the referees & editor(s) for their constructive comments and appreciation.

Response to Reviewer’s Comments

The authors are grateful to the reviewers for their suggestions that have all contributed to improving the manuscript. Once again, the authors are extremely thankful for the observations and the comments of the reviewers. All the comments are appropriately addressed and now the quality of the article has been appreciably enhanced before the consideration for publications. The rebuttal file is enclosed indicating the revisions incorporated in the article as suggested. The revisions are carried out in yellow colour in the text of the manuscript for better visibility to the reviewers and as well as to the editor. We have made the modifications as per their suggestions in the revised manuscript and changes are also marked up using the “Yellow font colour” function.

All in all, the authors should thank the reviewers for their meticulous observations in reviewing the article. All the issues raised by the authors are appropriately addressed as stated in the following table.

Authors’ Reply to The Editor and Reviewer 2

Title: A sustainable methodology using lean and smart manufacturing for the cleaner production shop floor management in industry 4.0

Reply to Editor: Authors are thankful to the Editor for providing constructive comments on our manuscript. The authors’ have incorporated all the suggestions given by the editor and reviewer. The authors’ have formatted the manuscript along with references as per journal guidelines.

Manuscript ID: Mathematics - 1523846

Comment 1 – Part 3, It would be better, if you could give common features and differences of Industry A and Industry B into the table and you have to justification. Why did you choose this industries?

Authors’ Reply – The authors are thankful to the reviewer for giving constructive comments, and the possible ground behind the selection of these industries for the present study because after contemplating from the prior literature studies, we get acquainted regarding the fact that both the Industries, Automobile, and Mining Machinery are relentlessly facing tremendous problematic concerns regarding enhance the operational excellence, yield efficiency, process efficiency, technological and organizational innovations, prevent production and quality failures, production-rate with superior quality of the finished products, financial profitability and optimize production processes by minimizing the non-value added activities and wastes during production flow-stream.

Additionally, in this regard, by leveraging Industry 4.0 and smart manufacturing with lean related high-tech advanced technological frameworks, the automotive and mining machinery manufacturers can easily address the processes-driven quality and throughout losses in production and assembly processes which includes, surface quality issues, coating issues, paint thickness problems, dashboard assembly issues, interiors and more, can all be alleviated.

Furthermore, Industry 4.0 uses digital technologies to make manufacturing more productive, efficient, agile, flexible, and responsive to customers. Thereby, it is able to create a smart factory where the advanced technologies work together to optimize the manufacturing system and improve customer satisfaction.

The current research aims to enhance production-rate within available resources and constraints. It has been observed that the production management teams of both industries were facing problems in achieving desired production enhancement within available resources and constraints. Therefore, the developed methodology has been implemented and achieved production enhancement within available resources and constraints to eliminate these problems.

As per instruction, the common characteristics and differences among the two industries A and B respectively have been detailed enumerated in the tabulated form as shown in the revised article. Furthermore, section 3 has significantly been improved.

The correction has been shown on page 7 lines 273-278, page 8, lines 314-317, lines 318-324, page 9, lines 325-330.

Comment 2 - Part 4 and 4.1 and 4.2 – Is it the Industry A (B) or the company A (B)?

Authors’ Reply – The authors are thankful to the reviewer for giving constructive comments.

Substantial modifications have been made on section 4, and the sub-sections, 4.1, and 4.2 respectively as shown in the revised manuscript.

The proposed methodology has been implemented with lean and smart manufacturing in industry A and industry B.

Where Industry A was the automobile industry and Industry B was the earthmoving machinery assembly unit.

The correction has been shown on pages 8-9 lines 318-330, page 9, line 333, and page 13, line 406.

Comment 3 Table 1 and Table 4 – Do you think, that the sample is representative?

Authors’ Reply – The authors are thankful to the reviewer for giving constructive comments.

As per the valuable suggestions, the authors would like to mention that the samples in Tables 1 and 4 respectively are utterly representative. Tables 1 and 4 were undoubtedly showing an unbiased reflection of the larger group. Moreover, in this regard, the data as shown in the tables are completely accurate and certainly reflects the specified characteristics of the larger dataset.

In the light of the aforementioned concern, the authors would like to mention that the findings of the present study have been inferred and concluded as per the defined objectives and the related outcomes have been affirmed with the prior literary studies to confirm the reliability of the results.

The correction has been shown on pages 18 lines 489-496, page 18, lines 502-517, and page 25, lines 545-557.

Comment 4 - Part 4.1.2. – As you write, the system uses push system. How do you want to implement your proposed methodology with Industry 4.0 usage, while most of the parts that uses Industry 4.0 are based on the pull principle?

Authors’ Reply – The authors are thankful to the reviewer for giving constructive comments; in the present work, the two case studies have been carried out for the automobile and earthmoving equipment manufacturing industries, respectively. Now, the automobile industry has used a push system, but the earthmoving industry has based on a pull system. The developed methodology has been implemented for both industries and obtained results have been evident that the productivity has been enhanced with high yield efficiency, process efficiency, technological and organizational innovations, prevent production and quality failures, within available resources.

It has also been observed from the result analysis for both industries that the overall improvement of production rate and operational excellence in the earthmoving machinery manufacturing industry is far better than the automobile industry.

In addition, the result has been revealed that the developed methodology could significantly provide a higher productivity rate in industry 4.0. 

The correction has been shown on pages 8 lines 318-330, and pages 25-27, lines 567-610.

Comment 5 - Table 2 and Table 5 – It would be better to replace these two tables with pictures – something like process maps.

Authors’ Reply – Authors are thankful to the reviewer for giving constructive comments, as per instructions, the process maps have been included in the revised manuscript.

The correction has been shown on page 13 lines 395-397, page 17, lines 458-459.

Comment 6 - Lines 420 – 433 – I think that is not right to give something like literature review into the section “Results and Discussions”. Please, find better place for this paragraph.

Authors’ Reply – Authors are thankful to the reviewer for giving constructive comments, and as per the instructions, Lines 420 – 433 has been improved.

In the light of the aforementioned valuable suggestion, the authors would like to mention that the content as mentioned on lines 420-433 has not actually resembled the Literature review, rather then, it apparently shows the comparative analysis of the results obtained from the current experimental study with the findings from the previous literature studies.

Furthermore, the authors would like to highlight that during this comparative evaluation, it was discovered that the contribution of the present research study are more remarkable and excellent.

The correction has been shown on page 25, lines 545-557.

A Scientific explanation of the obtained results has been refined and ameliorated upto fervent extent. Results are enumerated, methodology are utterly described, interpretation have been corelated with results and previous literature findings. The overall summary should indicate the progress of the research and the limitations. 

Note: All the necessary changes/added sentence has been shown by yellow colour.

Thank you very much in advance for taking your time in reviewing this manuscript.

Sincerely, we hope you will find our revision satisfactory.

Thanks, in anticipation.

Regards,

Shubham Sharma

Gianpaolo Di Bona

(Corresponding author)

Round 2

Reviewer 1 Report

I appreciate the authors for addressing my comments. 

Author Response

Thanks to Reviewer 1